# MaskMentor: Unlocking the Potential of Masked Self-Teaching for Missing Modality RGB-D Semantic Segmentation

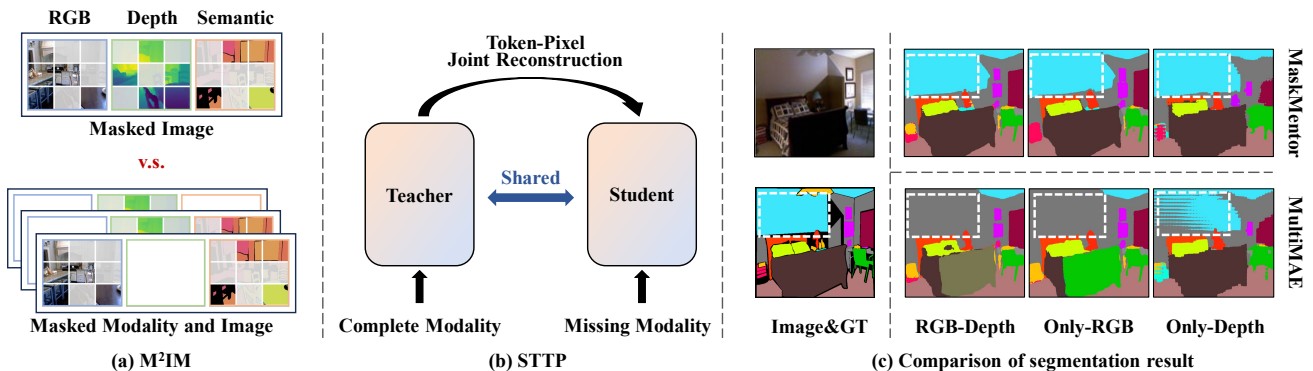

**Figure 1: High-level illustration of MaskMentor. (a) M²IM combines both modality- and patch-level random masking to enforce cross-modal prediction for modality-missing modeling. (b) STTP uses the teacher with complete modality input to supervise the student with modality missing input through joint token- and pixel-wise reconstruction, where the student and teacher share parameters. (c) MaskMentor delivers perceptually more accurate segmentation results under diverse modality-missing input conditions compared to the state-of-the-art method MultiMAE [1].**

## ABSTRACT

Existing RGB-D semantic segmentation methods struggle to handle modality missing input, where only RGB images or depth maps are available, leading to degenerated segmentation performance. We tackle this issue using **MaskMentor**, a new pre-training framework for modality missing segmentation, which advances its counterparts via two novel designs: **M**asked **M**odality and **I**mage **M**odeling (**M²IM**), and **S**elf-**T**eaching via **T**oken-**P**ixel Joint reconstruction (**STTP**). M²IM simulates modality missing scenarios by combining both modality- and patch-level random masking. Meanwhile, STTP offers an effective self-teaching strategy, where the trained network assumes a dual role, simultaneously acting as both the teacher and the student. The student with modality missing input is supervised by the teacher with complete modality input through both token- and pixel-wise masked modeling, closing the gap between missing and complete input modalities. By integrating M²IM and STTP, MaskMentor significantly improves the generalization ability of the trained model across diverse input conditions, and outperforms state-of-the-art methods on two popular benchmarks

by a considerable margin. Extensive ablation studies further verify the effectiveness of the above contributions.

## CCS CONCEPTS

• **Computing methodologies** → **Image segmentation**;

## KEYWORDS

Missing Modality, RGB-D Semantic Segmentation

## 1 INTRODUCTION

Semantic segmentation [7, 9, 38], as a fundamental and challenging problem in computer vision aims to predict the pixel-level categories for an input image, which has found wide applications in real scenarios. Compared to its single-modal (*i.e.*, with RGB input) counterparts [2], RGB-D segmentation integrates multi-modality input information for more precise segmentation results, and therefore has recently attracted increasingly more attention from the community.

Most existing approaches [34, 41, 42] address RGB-D segmentation by emphasizing the fusion of multi-modal features through carefully designed attention and fusion modules. Though superior performance has been achieved, they require that both RGB image and depth are available as input during inference, and can hardly generalize to missing modality cases where RGB or depth may be inaccessible (Figure 1 (c)). This is a common occurrence in practice due to hardware limitations, which significantly restricts the practical application of these approaches. Unfortunately, the problem of RGB-D segmentation with possible missing modalities has received less attention in the literature, leaving it largely underexplored.

To address the above issue, a recent study [1] makes one of the initial attempts by introducing a multi-modal pre-training strategy based on masked image modeling (MIM), which achieves effective feature alignment across modalities, yielding promising improvements in segmentation accuracy. Nevertheless, it is still limited in two aspects. First, the Multi-MAE proposed by [1] is pre-trained on complete input modalities. Consequently, although the utilization of MIM improves the alignment and generation of cross-modal information to a certain extent, it still faces challenges in achieving desirable segmentation results when dealing with modality-missing scenarios due to the input inconsistency between training and inference. Secondly, it only employs pixel-level masked modeling for pre-training, which overlooks the potential benefits of feature-level mask modeling. Recent research [5, 11, 27] indicates that utilizing tokenized semantic features can offer enhanced supervision for MIM. However, it remains uncertain whether this principle can be further extended to the RGB-D segmentation task with missing modalities.

In light of the above observation, we propose a new RGB-D missing modality segmentation paradigm called **MaskMentor** to unlock the potential of MIM, which consists of the following two unique designs. We first devise a **M**asked **M**odality and **I**mage **M**odeling (**M$^2$IM**) pre-taining approach, as shown in Figure 1(a), which extends the idea of MIM from image patch-level to modality-level. The modality-level masking will randomly mask out the entire input of one modality to mimic the missing modality scenario during inference. By combining both patch and modality masking, the pre-training target will force the network to reconstruct masked modality from a sparse set of unmasked patches of other modalities. As a result, the pre-trained network will not only learn to encode intra-modal information but also enforce its cross-modal predictive power, thereby significantly benefiting missing-modality segmentation. In addition, we further present **S**elf-**T**eaching via **T**oken-**P**ixel Joint Reconstruction (**STTP**) method for more effective training, as shown in Figure 1(b). Under the self-teaching framework with MIM, the trained network acts as both the teacher and the student simultaneously with shared parameters. The teacher is learned with complete modalities as input to perform pixel-wise reconstruction, whose output tokens will provide supervisory signals to enhance the student with missing modality input. By alternatively updating the teacher and student network, STTP incorporates fine-grained spatial characteristics and high-level semantic information from pixel- and token-wise supervision, respectively. As STTP does not train separate teacher and student networks, it permits complete modality input to improve missing modality input in a more cost-effective manner.

By integrating the aforementioned two techniques, MaskMentor significantly improves the effectiveness of MIM-based self-teaching pre-training, leading to more superior and robust RGB-D semantic segmentation with arbitrary missing modality input (See Figure 1 (c)). The main contributions of this work can be summarized into three folds.

- We propose the MaskMentor framework, which unlocks the potential of MIM for more accurate missing modality RGB-D segmentation.

- We design M$^2$IM pre-training approach, which combines both patch- and modality-level masking and significantly enforces the cross-modal modeling capabilities of MIM.
- We present STTP, a MIM-based self-teaching method, which can effectively improve the predictive power from missing modality input using supervisions offered by complete-modality data and integrates fine-grained spatial characteristics with high-level semantic information.

Experiments on two widely adopted benchmark datasets have verified the above contributions. Source code and pre-trained model will be made publicly available.

## 2 RELATED WORK

**RGB-D Semantic Segmentation.** Many existing RGB-D segmentation works [25, 40, 41] have shown promising results compared to single-modal semantic segmentation [13, 31, 38] by leveraging depth information. In the pursuit of the interaction and alignment between RGB and depth modalities, the dominant methods [34, 41, 42] focus on designing fusion modules to align and combine RGB and depth features. Though superior performance has been achieved, these methods require that both RGB image and depth are available as input during inference. However, this requirement restricts their applicability to situations commonly encountered in practice, where the RGB or depth modality may be unavailable.

**Missing Modality in Multi-modal Learning.** Perception with missing modalities has garnered growing attention in vision-text classification [19, 23], autonomous driving [39], *etc.* In the semantic segmentation field, some initial efforts have been made by recent works [1, 42]. Among them, [42] proposes a cross-modal fusion paradigm to address arbitrary modal segmentation, which tackles different modalities by training separate models. [1] is more correlated to ours, which proposes a cross-modal masked image modeling pre-training approach for modality missing RGB-D segmentation. Nonetheless, it is trained on complete input modalities, which limits its ability in handling modality missing input. Besides, it only employs pixel-level reconstruction for MIM and overlooks the potential of feature-level reconstruction.

**Masked Image Modeling.** MIM [3, 15] has become a predominant pre-training approach in computer vision. Prior methods [3, 15] mainly focus on the image modality and perform self-supervised learning by recovering the masked content from visible image patches, Recent works [1, 33] extend the MIM technique from image to multi-modal input, including language, depth, audio, *etc.* Meanwhile, other works [11] also explore to reconstruct tokenized semantic features for MIM, yielding more promising results.

**Self-training.** Self-training [18] is a special technique of knowledge distillation, which requires that the parameters-shared teacher and student be optimized simultaneously to transfer knowledge within the same model. Previous research has explored distilling the student model from the perspective of aligning logits output[24, 37] and intermediate representation[16, 17]. The latter attempts to optimize student by intimating the teacher at a more granular level, which may enable the student to learn richer and more profound knowledge. Based on this idea, work [43] transfer the knowledge from deeper portion of the networks to shallow layers to enhance

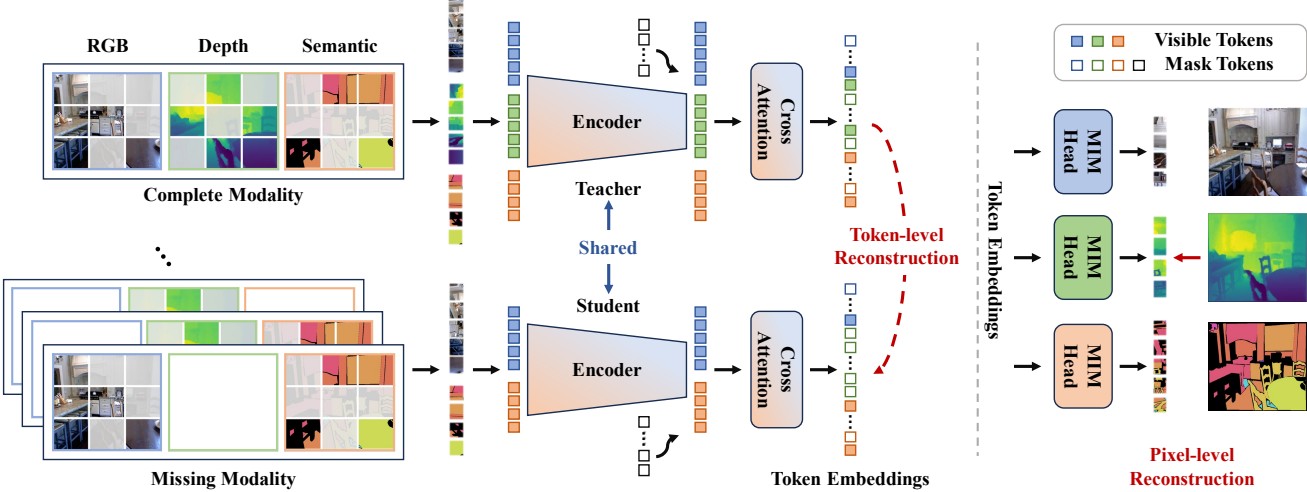

**Figure 2: Overview of the proposed MaskMentor framework in the pre-training stage. It consists of a Transformer encoder and multiple mask image modeling (MIM) head. The encoder serves as both a teacher and a student with shared parameters. The teacher receives complete modality data and performs pixel-level masked modeling. On the other hand, the student receives data that at least one modality is randomly masked and conducts modality-level masked modeling upon the remaining input modalities. During the self-teaching process, the teacher provides token-level knowledge of the missing modality to facilitate student learning.**

the overall performance of model, while recent research [29] brings closer the latent features of the same image under various data augmentations to align and unify visual semantics. Differently, our method employs token- pixel joint reconstruction in self-training manner to narrow down the intermediate representation between missing and complete modalities for gaining robust performance in any missing modality scenarios.

## 3 MASKMENTOR FOR RGB-D SEGMENTATION WITH MISSING MODALITIES

### 3.1 Problem Setting

In this paper, we investigate the task of RGB-D semantic segmentation with missing modalities. Specifically, we are given the complete-modality data including both RGB images and depth maps as input to train a semantic segmentation model. During testing, the input modalities may be arbitrarily missing, *i.e.*, either the complete-modality input is provided, or only a single modality is given with the other modality missing. This missing modality setting is closely aligned with real scenarios but presents a more formidable challenge compared to conventional RGB or RGB-D segmentation. As the input involves multi-modal data and may be inconsistent between training and testing, the trained model should be able to not only harness the advantage of multi-modal input but also well tackle the training-testing input discrepancy.

A straightforward idea to address RGB-D segmentation with missing modalities is to train separate models corresponding to different input modalities. During testing, the system should select a specific model for inference according to the input modalities. However, this will linearly increase the training complexity and the memory consumption of model deployment. Instead, this paper

proposes a novel framework called MaskMentor, which allows training a single model to unify different input cases, giving rise to a more elegant alternative to solving the aforementioned challenges.

### 3.2 Overview

Our proposed MaskMentor consists of a pre-training and a fine-tuning stage. Figure 2 presents an architectural overview of the pre-training stage, during which we train a Transformer network by following the principle of multi-model MIM with self-teaching. The pre-trained transformer network comprises an encoder and multiple MIM heads corresponding to different modalities. During fine-tuning, MIM heads will be discarded. A randomly initialized decoder is introduced after the pre-trained encoder and the entire network will be fine-tuned for RGB-D segmentation with missing modalities. The key designs of MaskMentor include Masked Modality and Image Modeling ($M^2IM$) and Self-teaching via Token-Pixel Joint Reconstruction (STTP), whose details will be further explained in the following.

### 3.3 Masked Modality and Image Modeling

Masked Image Modeling [3, 15] has been proven to be an effective self-supervised learning approach that randomly masks out input image patches and trains a network to restore these masked patches from visible ones. Recently, this idea has been successfully transferred to multi-modal input cases by [1]. To enforce cross-modal modeling, [1] improves the random masking manner through a newly developed multi-modal token sampling approach to ensure a more diverse sampling of visible tokens from different modalities. Although the trained model is more capable of cross-modal prediction, its potential against missing input modalities is largely

---

**Algorithm 1** Two-stage multi-modal data masking in M$^2$IM.

---

**Input:** Data of $K$ modalities $Q = \{X_k | k = 1, 2, \ldots, K\}$, modality masking probability $p_m$.

**Output:** Masked data $O$.

1: Initialize $O = \emptyset$, $g = 0$.
2: Randomly sort input data $Q \leftarrow \text{RandomSort}(Q)$.
3: **for** $k = 1, 2, \ldots, K - 1$ **do**
4:     Uniformly sample $v$ from $[0, 1]$.
5:     **if** $v >= p_m$ **then**
6:         $O \leftarrow O \cup \{X_k\}, g \leftarrow 1$.
7:     **end if**
8: **end for**
9: **if** $g == 0$ **then**
10:     $O \leftarrow O \cup \{X_K\}$
11: **else**
12:     Uniformly sample $v$ from $[0, 1]$.
13:     **if** $v >= p_m$ **then**
14:         $O \leftarrow O \cup \{X_K\}$
15:     **end if**
16: **end if**
17: **for** each $X$ in $O$ **do**
18:     $X \leftarrow \text{PatchMasking}(X)$
19: **end for**=0

---

restricted as the pre-training process of [1] is still performed on complete input modalities.

To remedy this deficiency, our proposed M$^2$IM employs a two-stage masking strategy, combining the modality- and image patch-level masking. Detailed procedure is illustrated in Algorithm 1. The first stage (Line 2-16) performs modality-level masking, where all image patches of a masked modality will be entirely discarded. Specifically, we are given input data from $K$ modalities. For the first $K - 1$ modalities, we mask each modality by a probability of $p_m$. For the last modality, if all the first $K - 1$ modalities are masked out, it will be preserved (Line 10). Otherwise, it will be masked by the same probability of $p_m$. This implementation can avoid the case where all the $K$ input modalities are masked out. However, the mask probabilities of the first $K - 1$ and the last modalities are not equivalent. Therefore, we randomly sort the $K$ modalities each time before the above masking process to achieve the balance between input modalities. After the modality masking stage, there will be $M$ unmasked modalities remaining with $1 \leq M \leq K$. The second stage then applies image patch masking to the remaining $M$ modality (Line 17-19) following the same routine of [15].

After the above masking process, all the visible patches of unmasked modalities are tokenized via separate projection layers, concatenated, and then passed through the Transformer encoder. Following [15], mask tokens are inserted into the output token sequence of the encoder, serving as placeholders for the masked patches. Both masked and visible tokens are further fed into a cross-attention module to perform interaction and produce the output token embeddings. Finally, modality-specific MIM heads take these output embeddings as input to reconstruct masked patches of all modalities. By using the two-stage masking strategy, M$^2$IM explicitly mimics the missing modality cases during inference and forces the trained model to better generalize across diverse input situations.

## 3.4 Self-Teaching via Token-Pixel Joint Reconstruction

The aforementioned M$^2$IM technique only adopts the pixel-level reconstruction target for training, while recent evidence [11] suggests that using token reconstruction for MIM can deliver more high-level and abstract information. We aim to investigate whether these two types of reconstruction targets are mutually complementary in the multi-modality scenario. The first challenge we encounter is how to obtain the target tokens. For this purpose, we propose a new pre-training framework called Self-Teaching via Token-Pixel Joint Reconstruction (STTP), which is built upon M$^2$IM technique.

As shown in Figure 2, the pre-trained model simultaneously acts as the teacher and student under the STTP framework. The teacher is trained with multi-modal MIM [1], where the input data is from complete modalities, and data masking is only performed on the patch level. The teacher learns to reconstruct the masked patches from visible input ones. In comparison, the student is trained in the M$^2$IM style with input data of missing modalities which has been masked using the proposed two-stage masking approach. The student learns to reconstruct both the masked patch of all modalities as well as the token embeddings produced by the teacher (See Figure 2). For each input batch during training, we first train the teacher network for three iterations and then train the student for one iteration.

The proposed STTP offers two key advantages. First, using the teacher that receives complete modality input to supervise the student with missing modality input can significantly narrow down the performance gap between various input conditions. Second, STTP combines the principles of knowledge distillation with M$^2$IM, and inherently marries the advantages of both pixel- and token-level reconstruction. In addition, STTP under the self-teaching framework eliminates the need for training separate teacher and student networks, giving rise to a more cost-effective pre-training method. As shown in our experiments, STTP effectively benefits the downstream missing-modality RGB-D segmentation task.

## 3.5 Overall Training Pieline

During the pre-training stage of our MaskMentor framework, we exploit $K = 3$ input modalities, including RGB images, depth maps, and semantic segmentation maps, where depth maps characterize the geometric information and segmentation maps encode the scene semantics. The network is warmed up for around 100 epochs by training on the MIM task using complete input modalities, and then trained with the proposed STTP approach for another 400 epochs. Cosine similarity is adopted to measure the token-level reconstruction loss while the pixel-level reconstruction loss follows the implementation of [1]. During fine-tuning, And the MIM head is replaced with a randomly initialized ConNeXt [21] decoder. The entire network is trained for RGB-D segmentation with missing modalities for 500 epochs.

## 4 EXPERIMENTS

### 4.1 Setting Up

**Dataset.** We perform experiments on two widely adopted RGB-D semantic segmentation datasets, including NYUDepthV2 [28]

**Table 1: Performance comparison for RGB-D segmentation on NYUDepthV2 [28] and SUN RGB-D [30].**

| Method | NYUDepthV2 | | SUN RGB-D | |
|---|---|---|---|---|
| | mIoU | mAcc | mIoU | mAcc |
| FuseNet [14] | 37.9 | 50.4 | 37.3 | 48.3 |
| RDFNet [26] | 50.1 | 62.8 | 47.7 | 60.1 |
| SSMA [32] | 48.7 | 60.5 | 45.7 | 58.1 |
| AsymFusion [36] | 51.2 | 64.0 | - | - |
| SA-Gate [8] | 52.4 | 64.8 | 49.4 | 61.3 |
| CEN [35] | 52.5 | 65.0 | 51.1 | 63.2 |
| SGNet [6] | 51.1 | 63.1 | 48.6 | 60.9 |
| ShapeConv [4] | 51.3 | 63.5 | 48.6 | 59.2 |
| Omnivore [12] | 54.0 | - | - | - |
| TokenFusion [34] | 54.2 | 66.9 | 53.0 | 64.1 |
| MultiMAE [1] | 56.8 | 69.9 | 51.5 | 63.2 |
| PGDENet [44] | 53.7 | 66.7 | 51.0 | 61.7 |
| CMX [41] | 56.9 | - | 52.4 | - |
| CMXNeXt [42] | 56.9 | - | 51.9 | - |
| **MaskMentor** | **57.9** | **70.4** | **53.0** | **66.4** |

**Table 2: Performance comparison for missing modality segmentation on NYUDepthV2 [28] and SUN RGB-D [30]. "Only-RGB" and "Only-Depth" refer to the input scenarios where either the RGB image alone or depth map alone is available as the provided modality, respectively.**

| Dataset | Method | Only - RGB | | Only - Depth | |
|---|---|---|---|---|---|
| | | mIoU | mAcc | mIoU | mAcc |
| NYUDepthV2 | RefineNet [20] | 46.5 | 59.0 | 34.3 | 45.6 |
| | CEN [35] | 39.6 | 51.8 | 19.3 | 29.0 |
| | TokenFusion [34] | 50.6 | 63.3 | - | - |
| | CMX [41] | 46.7 | 61.0 | - | - |
| | MAE [15] | 50.8 | - | 23.4 | - |
| | MultiMAE [1] | 52.1 | 65.9 | 41.6 | 51.3 |
| | CMNeXt [42] | 52.2 | 66.2 | 33.5 | 42.4 |
| | **MaskMentor** | **53.3** | **66.5** | **44.0** | **56.8** |
| SUN RGB-D | RefineNet [20] | 47.0 | 57.7 | - | - |
| | TokenFusion [34] | 48.1 | 61.3 | - | - |
| | MultiMAE [1] | 48.3 | 61.9 | 40.0 | 48.6 |
| | **MaskMentor** | **49.8** | **63.1** | **41.2** | **49.1** |

and SUN RGB-D [30]. The NYUDepthV2 dataset [28] consists of 1449 RGB-Depth image pairs with 40 distinct categories of indoor objects. The training set comprises 795 image pairs, while the test set includes 654 pairs. All images in this dataset are of size $480 \times 640$ pixels. The SUN RGB-D dataset [30] consists of 10,335 real RGB-D pairs representing room scenes, and it contains a total of 37 object categories. The training set comprises 5,285 pairs, while the testing set has 5,050 pairs. Each image in this dataset has a resolution of $730 \times 530$ pixels. During training, we apply data augmentation techniques, including random flipping, cropping, and rescaling following the approach described in [1].

**Implementation.** We adopt ViT-B [10] as the encoder, while other network parameters are all randomly initialized. The learning rate of the pre-training stage is initially set to $1e-5$ and the cosine learning rate schedule is employed. The AdamW [22] optimizer is used with a batch size of 12. The fine-tuning stage adopts an initial learning rate of $3e-5$ with a cosine learning rate schedule and a batch size of 2. The mask rate $p_m$ at the modality level is

uniformly set to 0.5. For a fair comparison against existing methods, we keep the other implementations consistent with [1], including input resolution, patch size, patch-level masking ratio, positional embedding, *etc*.

**Evaluation Metrics.** Following the previous works [34, 41], we utilize two evaluation metrics for quantitative assessment of the segmentation results, including mean Accuracy (mAcc) which offers an overall measure of the model's classification capability, and mean Intersection over Union (mIoU) that is to measure the average intersection over union across all categories.

## 4.2 Overall Comparison

We perform comprehensive evaluations of our proposed method against state-of-the-art methods for both complete (i.e., RGB-D) and missing modalities (i.e., only RGB and only Depth) semantic segmentation. It is worth noting that, unlike the compared methods that individually train separate models for different input modalities,

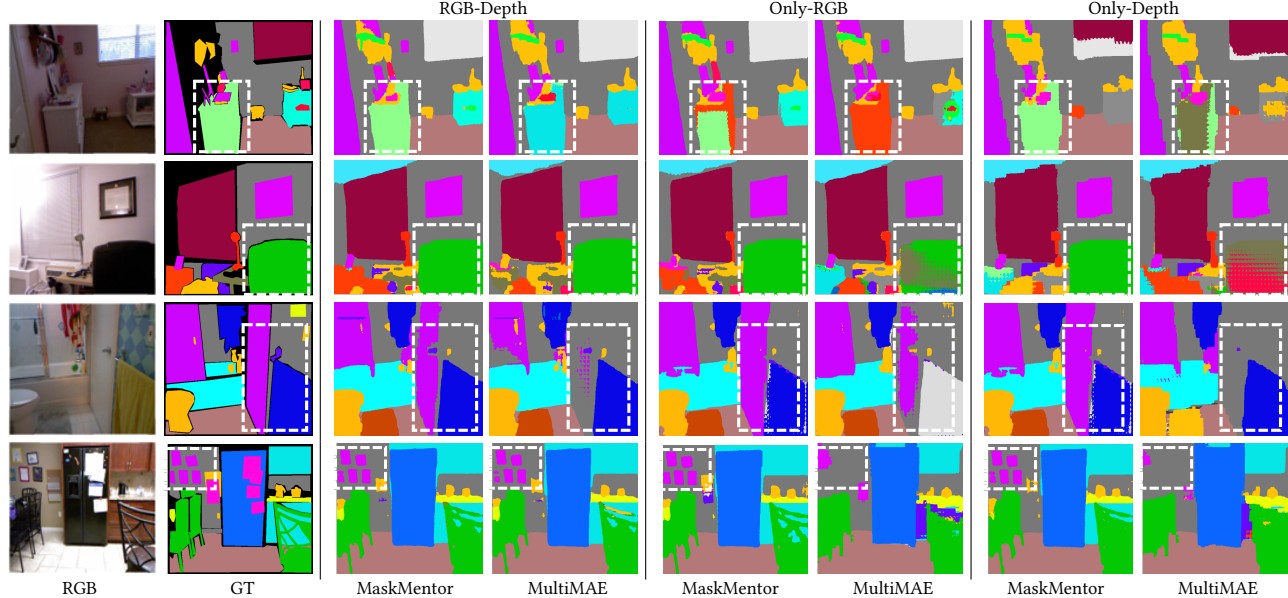

**Figure 3: Visual comparison of our MaskMentor and MultiMAE [1] on semantic segmentation performance across various scenes in the NYUDepthV2 test set [28].**

our approach uses the same trained model to test different input modality scenarios.

**Complete Modality Segmentation Performance.** Table. 1 reports the quantitative evaluation for RGB-D semantic segmentation on NYUDepthV2 and SUN RGB-D test datasets. The proposed Mask-Mentor achieves consistently superior performance compared to the existing methods that are specifically trained for the RGB-D segmentation task. Particularly, it is noteworthy that our MaskMentor outperforms the recent best method CMXNeXt [42] by 1.8% and 2.1% in terms of mIoU on NYUDepthV2 and SUN RGB-D datasets, respectively. Moreover, compared to MutliMAE [1] that employs the MIM for network pertaining, MaskMentor also shows significant superiority on both datasets. These results indicate that our Mask-Mentor can effectively learn the multi-modal image representations for the downstream semantic segmentation task.

**Missing Modality Segmentation Performance.** We further evaluate the segmentation performance of the models when they receive only the RGB image ("Only-RGB") or depth map ("Only-Depth") as input. Results are provided in Table 2. The results indicate that our MaskMentor exhibits substantial advantages in both two modality missing scenarios on the test datasets, even though the compared methods are specifically trained for individual modalities. It is particularly noteworthy that our method achieves significant improvements even when the RGB modality is missing, outperforming the compared methods.

**Segmentation Visualization.** Figure 3 provides qualitative segmentation results of the proposed MaskMentor and MultiMAE [1]. It can be observed that MaskMentor is capable of consistently recognizing more accurate object categories under different modality input scenarios, highlighting the robustness of our method in addressing the challenge of modality absence.

**Table 3: Ablation studies on the proposed $M^2IM$ and STTP.**

| Dataset | Method | RGB-Depth | | Only - RGB | | Only - Depth | |
|---|---|---|---|---|---|---|---|
| | | mIoU | mAcc | mIoU | mAcc | mIoU | mAcc |
| NYUDepthV2 | baseline | 56.0 | 68.7 | 46.7 | 56.3 | 33.8 | 44.2 |
| | + MIM | 56.8 | 69.9 | 47.2 | 59.2 | 38.9 | 50.6 |
| | + $M^2IM$ | 56.8 | 70.0 | 52.0 | 65.9 | 42.5 | 55.2 |
| | + $M^2IM$ + STTP | **57.9** | **70.4** | **53.3** | **66.5** | **44.0** | **56.8** |
| SUN RGB-D | baseline | 50.0 | 61.9 | 42.2 | 53.3 | 31.6 | 42.1 |
| | + MIM | 51.5 | 63.2 | 44.4 | 54.6 | 36.1 | 44.3 |
| | + $M^2IM$ | 52.0 | 64.7 | 48.5 | 61.6 | 37.8 | 47.9 |
| | + $M^2IM$ + STTP | **53.0** | **66.4** | **49.8** | **63.1** | **41.2** | **49.1** |

## 4.3 Ablation Study

We design various ablation studies to evaluate the effectiveness of our core contributions. Unless otherwise specified, all experiments are conducted using the default training configurations as described in Section 4.1.

**Effectiveness of $M^2IM$ and STTP.** To verify the effectiveness of $M^2IM$ and STTP, three variants are proposed as shown in Table 3. The "baseline" refers to the model that undergoes direct fine-tuning on the segmentation task without pre-training, which performs inferior particularly in scenarios where only RGB or Depth is available as input. When adding the MIM-based pertaining as MultiMAE [1], the performance is improved. Introducing our proposed $M^2IM$ results in significant performance improvements compared to the MIM-based variant. Specifically, it achieves mIoU improvements of 10.2% and 9.2% for the Only-RGB and Only-Depth settings on the NYUDepthV2 dataset, respectively. Similarly, on the SUN RGB-D dataset, it achieves mIoU improvements of 12.1% and 19.62% for the Only-RGB and Only-Depth settings, respectively. These results

**Table 4: More ablation studies of STTP in terms of self-teaching and network supervision on NYUDepthV2 dataset [28]. "KD" refers to Knowledge Distillation, where a separately pre-trained teacher model distills its learned knowledge to guide the student model.**

| Method | RGB-Depth | | Only - RGB | | Only - Depth | |
|---|---|---|---|---|---|---|
| | mIoU | mAcc | mIoU | mAcc | mIoU | mAcc |
| Pixel-level MIM | 56.8 | 70.0 | 52.0 | 65.9 | 42.5 | 55.2 |
| Token-level MIM | 56.6 | 69.2 | 50.4 | 63.3 | 39.5 | 50.3 |
| KD+M$^2$IM | 56.8 | 69.3 | 50.7 | 65.1 | 43.1 | 55.8 |
| MaskMentor | **57.9** | **70.4** | **53.3** | **66.5** | **44.0** | **56.8** |

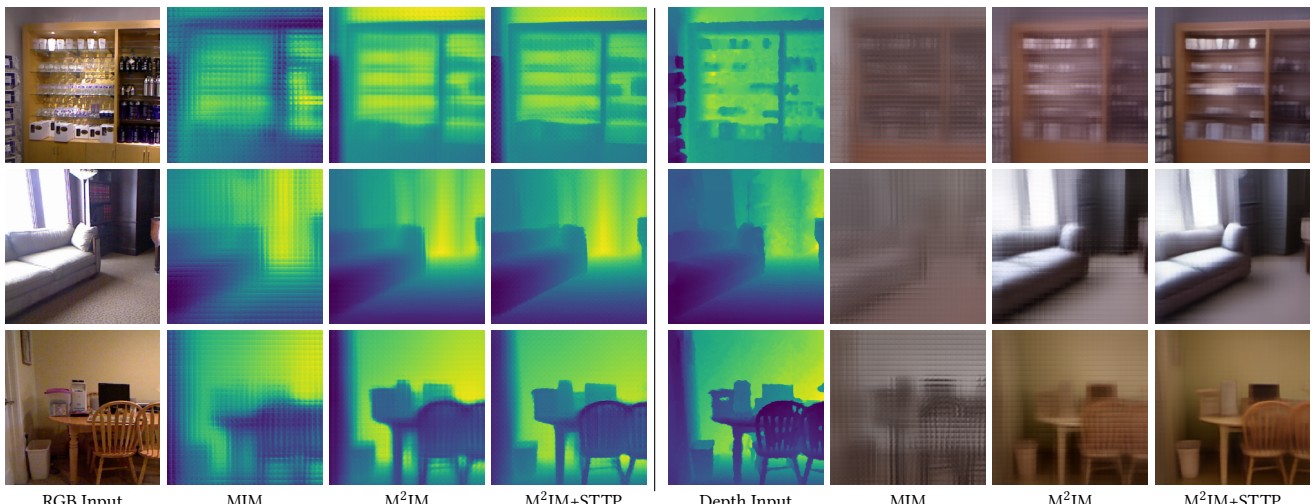

RGB Input     MIM     M$^2$IM     M$^2$IM+STTP     Depth Input     MIM     M$^2$IM     M$^2$IM+STTP

**Figure 4: Visualization of the reconstructed modality. Given the RGB or Depth image as input, the pre-trained model equipped with the proposed M$^2$IM+STTP can produce the other modality (*i.e.*, depth map or RGB image) with more plausible information.**

indicate that the cross-modal masked modeling by the proposed M$^2$IM provides better alignment in missing-modality scenarios. Additionally, the integration of STTP further leads to a considerable improvement in segmentation performance, demonstrating the effectiveness of our self-teaching strategy with token-pixel joint reconstruction.

**Separate Teacher-Student *v.s.* Self-Teaching.** We take a further step to investigate the critical factors of STTP. We first evaluate the impact of self-teaching and design a variant where the teacher is separately trained and then provides supervision for the student. As indicated by the last two rows in Table 4, our parameter-shared teacher-student strategy achieves better performance, while also making our method cost-effective.

**Effectiveness of Token-Pixel Reconstruction.** Our network is trained with the supervision of token-pixel joint reconstruction. To quantitatively evaluate their contributions, we conducted ablation experiments. Comparing the results in the first two rows versus the last row of Table 4, it demonstrates that both pixel-level and token-level reconstruction are essential in improving the overall performance.

**Visualization of Modality Reconstruction.** Figure 4 provides a comprehensive illustration of the cross-modal reconstruction capabilities exhibited by the models that have undergone pre-training utilizing a variety of methodologies, including MIM, M$^2$IM, and M$^2$IM+STTP, respectively. Given either the RGB or depth map as input, our method (M$^2$IM+STTP) shows strong capability in generating the other modality with more plausible details.

## 5 CONCLUSION

In this paper, we introduce a novel framework named MaskMentor to address the challenging task of missing modality RGB-D semantic segmentation. Our method extends the idea of MIM from the image patch level to the modality level and forces the network to reconstruct the masked modalities from the visible ones, thus enhancing the model's capability of dealing with modality-missing situations. In addition, it incorporates token- and pixel-wise supervision under the self-teaching paradigm, where the student with missing modality input is supervised by the teacher with complete modality input. As such, fine-grained spatial characteristics and high-level information of multi-modal data are effectively integrated and the performance gaps of the model with diverse modality missing input conditions are further closed up. Extensive experiments on benchmark datasets verify the effectiveness of the proposed method. In our future work, we will extend our exploration to encompass a wider range of data modalities, such as language, audio, *etc.*

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
