# OpenReview forum: "MaskMentor: Unlocking the Potential of Masked Self-Teaching for Missing Modality RGB-D Semantic Segmentation"
_acmmm.org/ACMMM/2024/Conference — MM2024 Poster_

### Official Review · Reviewer_qxxK · 2024-05-13

**Rating:** 5
**Confidence:** 3

**Summary:**

The manuscript unlocks the potential of MIM for missing modality RGB-D segmentation. M2IM combines both patch- and modality-level masking and significantly enforces the cross-modal modeling capabilities of MIM. STTP uses the teacher with complete modality input to supervise the student with modality missing input through joint token- and pixel-wise reconstruction.

**Strengths:**

1. The method introduces M2IM which benefits missing modality RGB-D segmentation.
2. The method conducts a pre-training which combines both patch- and modality-level masking inspired by MultiMAE.

**Limitations:**

The problems include:
1. “multi-model MIM” in section 3.2 should be “multi-modal MIM”.
2. The details are not clear. In Figure 2, there are three groups of tokens fed into teacher’ encoder, while there are only two groups of tokens fed into student encoder. Moreover, the two encoders are shared. How to solve the problems with inconsistent length of input tokens?
3. Experiment is not adequate. The goal of the pretraining is to obtain a better pretraining parameters of encoder. DFormer[1] pretrained an RGB-D backbone for semantic segmentation task. Please give the comparison to verify the effectiveness M2IM for missing modal semantic segmentation.

References:
[1] Yin B, Zhang X, Li Z Y, et al. DFormer: Rethinking RGBD Representation Learning for Semantic Segmentation[C]//The Twelfth International Conference on Learning Representations. 2023.

**Suitability:**

2

---

### Official Review · Reviewer_TZaP · 2024-05-20

**Rating:** 3
**Confidence:** 4

**Summary:**

This paper addresses the challenges faced by existing RGB-D semantic segmentation methods in handling modality missing input by leveraging richer modality information. Specifically, it proposes MaskMentor, a novel pre-training framework for modality missing segmentation, and improves upon existing methods through two innovative designs.

**Strengths:**

1.A new pipeline is proposed.
2.Two innovative designs, Masked Modality and Image Modeling (M2IM) and Self-Teaching via Token-Pixel Joint reconstruction (STTP), are introduced to improve the corresponding methods. M2IM simulates modality missing scenarios by combining random masking at the modality and patch levels.
3.A self-distillation mechanism is proposed.

**Limitations:**

1.Is the loss in performance in scenarios with missing modalities due to the model not being retrained?

2.Are there any experiments conducted using only Depth images as input for the segmentation task?

The paper is logically coherent, but it lacks core experiments. If my doubts are addressed, it might change my perspective.

**Suitability:**

2

---

### Official Review · Reviewer_YU2F · 2024-05-25

**Rating:** 3
**Confidence:** 3

**Summary:**

This paper proposes a RGB-D sematic segmentation method with the target of handling modality missing input. Two key components are proposed, M2IM simulates modality-missing scenarios by combining both modality- and patch-level random masking. Meanwhile, STTP offers an effective self-teaching strategy, where the trained network assumes a dual role, simultaneously acting as both the teacher and the student.

**Strengths:**

The paper is overall well-written. The key ideas are easy to understand. Visual comparisons are good.

**Limitations:**

The paper is somewhat incremental to MultiMAE by introducing modality masking and Siamese structure driven token reconstruction loss. I have several concerns here:
1) Comparing M2IM with MIM, I think the contribution is more about conception rather than the technical part, since the modality masking may be regarded as a special case of patch masking by patch masking the entire image of certain modality images. Therefore, the multiMAE can also be applied to handle the modality missing cases as done in this paper. Also, in multiMAE, the authors do model testing with modality missing. Therefore, I think it is unfair to emphasize that “all” the compared methods in the experiments trained separate models for testing as in Line 579 of Page 5 and Lines 626 to 627 of Page 6 in the manuscript. Nevertheless, if the authors do training separate models for different modalities using multiMAE, I think it would be another unfair scenario as it would fail to unleash the power of multiMAE.
2) In sec 3.5, it says the Maskmentor is trained with 3 modalities. In Table 2, it says “Only-RGB”, “Only-Depth”, I’m wondering if the models tested are trained with 2 modalities ( semantic segmentation is always used?) or pre-trained with 3 modalities and fine-tuned with a single modality?
3) What is the difference in training strategies for Table 2 and Table 3 in the case of single modality, i.e., the “Only-RGB” and “Only-Depth”. Why did the indices differ so much for MultiMAE in Table 2 and MIM in Table 3?
4) How are the indices of different methods in Table 1 and Table 2 obtained? Are they borrowed from the original paper or re-trained using the same setting？ How did you ensure a fair comparison?
5) In Table 3, for the case of RGB-Depth, why the performance of MIM and M2IM are very close on NYUDepthV2 but on the SUN RGB-D M2IM shows more advantages?

Overall, I think the paper presents some good points but needs further clarification.

**Suitability:**

3

---

### Meta-Review · Area_Chair_Ex9H · 2024-07-01

**Recommendation:** Accept (Poster)
**Confidence:** 4

**Metareview:**

This paper proposes a RGB-D sematic segmentation method with the target of handling modality missing input. Two key components are proposed, M2IM simulates modality-missing scenarios by combining both modality- and patch-level random masking. Meanwhile, STTP offers an effective self-teaching strategy, where the trained network assumes a dual role, simultaneously acting as both the teacher and the student.

After the rebuttal, all reviewers are positive about the submission.